# Pacemaker translocations and power laws in 2D stem cell-derived cardiomyocyte cultures

**Christopher S. Dunham**[1]*, **Madelynn E. Mackenzie**[2], **Haruko Nakano**[3], **Alexis R. Kim**[1], **Michal B. Juda**[4], **Atsushi Nakano**[3,4,5,6,7], **Adam Z. Stieg**[8,9]*, **James K. Gimzewski**[1,8,9]

**1** Department of Chemistry and Biochemistry, University of California, Los Angeles, California, United States of America, **2** Department of Microbiology, Immunology & Molecular Genetics, University of California, Los Angeles, California, United States of America, **3** Department of Molecular, Cell and Developmental Biology, University of California, Los Angeles, California, United States of America, **4** Molecular Biology Institute, University of California, Los Angeles, California, United States of America, **5** Eli and Edythe Broad Center of Regenerative Medicine and Stem Cell Research, University of California, Los Angeles, California, United States of America, **6** Division of Cardiology, Department of Medicine, University of California, Los Angeles, California, United States of America, **7** Department of Cell Physiology, The Jikei University, Tokyo, Japan, **8** California NanoSystems Institute, University of California, Los Angeles, California, United States of America, **9** International Center for Materials Nanoarchitectonics (MANA), National Institute of Materials Science, Tsukuba, Japan

* csdunham@chem.ucla.edu (CSD); stieg@cnsi.ucla.edu (AZS)

**Data Availability Statement:** The data is available through Dryad at this doi: (https://doi.org/10.5068/D1PD72).

## Abstract

Power laws are of interest to several scientific disciplines because they can provide important information about the underlying dynamics (e.g. scale invariance and self-similarity) of a given system. Because power laws are of increasing interest to the cardiac sciences as potential indicators of cardiac dysfunction, it is essential that rigorous, standardized analytical methods are employed in the evaluation of power laws. This study compares the methods currently used in the fields of condensed matter physics, geoscience, neuroscience, and cardiology in order to provide a robust analytical framework for evaluating power laws in stem cell-derived cardiomyocyte cultures. One potential power law-obeying phenomenon observed in these cultures is pacemaker translocations, or the spatial and temporal instability of the pacemaker region, in a 2D cell culture. Power law analysis of translocation data was performed using increasingly rigorous methods in order to illustrate how differences in analytical robustness can result in misleading power law interpretations. Non-robust methods concluded that pacemaker translocations adhere to a power law while robust methods convincingly demonstrated that they obey a doubly truncated power law. The results of this study highlight the importance of employing comprehensive methods during power law analysis of cardiomyocyte cultures.

## Introduction

Recent investigations of human induced pluripotent and embryonic stem cell-derived cardiomyocytes (hiPSC-CM and hESC-CM, respectively) have shone significant light on numerous factors contributing to the development of the heart [1–6]. However, the mechanisms

**Funding:** This work was funded by grants from the National Institutes of Health (https://www.nih.gov/grants-funding). Grant R21HL124503 was awarded to AZS, JKG, and AN. Grants R01 HL142801 and HL146159 were awarded to AN. MBJ was supported by the UCLA Center for Duchenne Muscular Dystrophy Ruth L. Kirschstein National Research Service Award T32AR065972 'Muscle Cell Biology, Pathophysiology, and Therapeutics' from the National Institute of Arthritis and Musculoskeletal and Skin Diseases (https://grants.nih.gov/grants/guide/pa-files/pa-21-048.html). The funders had no role in study design, data collection and analysis, decision to publish, or preparation of the manuscript.

**Competing interests:** The authors have declared that no competing interests exist.

underlying the full maturation of cardiomyocytes to robust adult phenotypes remain unknown [1, 7, 8]. Phenotypic traits which have thus far failed to match those in adult cardiomyocytes include electrical impulse propagation, mechanical properties including sarcomere length and contractility, cell morphology, and gene expression [1, 9–11]. Additionally, some studies have suggested that defective cardiomyocyte development may play a role in a number of disease states such as cardiomyopathy and late myocardial dysfunction [2, 12, 13]. The inability to mature beyond the late fetal phenotype stage considerably limits applications for stem cell-derived cardiomyocytes in drug screening, disease modeling, and regenerative medicine [1, 2, 7, 14, 15].

Among the mechanisms which must be understood in the context of cardiomyocyte development is the establishment of pacemaker cells. These pacemaker cells are responsible for maintaining the rhythmic beating of all cells in the cardiomyocyte syncytium by means of action potential generation [16, 17]. The heart has a dedicated region of pacemaker cells, known as the sinoatrial node (SAN), that is responsible for maintaining a consistent beat rhythm throughout the lifetime of the organism [1, 18]. Many types of arrhythmias result from the disruption of rhythm-maintaining electrical impulses in the SAN, throwing the system into disarray [2, 19]. Although SAN is responsible for initiating heartbeat in the postnatal heart in physiological condition, all cardiomyocytes in the early embryonic heart are capable of generating autonomic beats. The pacemaker cells are specialized during mid-gestational stages. *In vitro* differentiation of hESC-CMs/hiPSC-CMs recapitulates this process [20]. Understanding how pacemaker regions arise and the role of pacemaker instability during cardiomyocyte maturation could provide insight into the development of the SAN and would help in furthering the current understanding of arrhythmias.

Difficulty in elucidating the mechanisms responsible for pacemaker development may be partially attributed to gaps in knowledge about the underlying interactions between cells in the cardiomyocyte culture (i.e. interactions within the cardiomyocyte network). Information in the form of environmental, physical, genomic, and chemical cues concerning the establishment of specialized cellular structures (e.g. intercalated discs) and functional roles (e.g. pacemaker cells, late fetal proliferating cells, ventricular conduction system-like cells) needs to be transmitted to the cardiomyocytes within the network [2, 21–24]. Insight into cardiomyocyte network dynamics and the processes responsible for information transfer may be attainable through the analysis of observable network characteristics (e.g. beat rate, pacemaker behavior, and biomechanical properties) for adherence to power laws.

Power laws—probability distributions of the form $p(x) \propto x^{-\alpha}$ –are of interest to several scientific fields because they provide important information about the dynamics of the system, e.g. long-range correlations, scale invariance, and self-similarity [25–27]. Several studies have explored whether power laws apply to cardiac systems, particularly in the context of tissue or cellular dysfunction, e.g. mitochondrial oxidative stress and arrhythmias [28–32]. Prior studies demonstrated power law behavior in investigations of heart (beat) rate variability, calcium load, and contractile stress in cardiomyocytes [33–37]. Studies have shown that aberrant cardiac systems exhibit a discernible change in the exponent of the power law measured for heart rate variability, which describes small variations in the interval between heart beats, in patients with myocardial infarction and coronary heart disease, and in heart transplant patients [29, 38–41]. Other studies focused on power laws as they relate to the mitochondrial network and the effect of oxidative stress on both the network and the dependent cardiac myocytes.

While intriguing, the applied methodologies in these studies fail to meet the standards defined in other fields, including: condensed matter physics, geology, and neuroscience, where power law analysis is more established [42–45]. In these fields, power laws are typically evaluated through a combination of methods, including: 1) calculation of the power law exponent,

α, via maximum likelihood estimation (MLE), 2) statistical assessments of how well the data fit to a proposed distribution using the Kolmogorov-Smirnov goodness-of-fit test, 3) log-likelihood ratio tests between power law (Eq 1), exponential (Eq 2), and other heavy-tailed distributions (i.e. distributions in which the tail probability decays polynomially rather than exponentially), including log-normal (Eq 3), Weibull (stretched exponential, Eq 4), doubly truncated power law (Eq 5), and other candidate distributions to determine which one demonstrates a superior fit to the data [42, 46–48].

$$f(x) = x^{-\alpha} \tag{1}$$

$$f(x) = e^{-\lambda x} \tag{2}$$

$$f(x) = \frac{1}{x} \star exp\left[\frac{(ln(x) - \mu)^2}{2\sigma^2}\right] \tag{3}$$

$$f(x) = (x\lambda)^{\beta-1} \star e^{-(\lambda x)^\beta} \tag{4}$$

$$f(x) = x^{-\alpha} \star e^{-\lambda x} \tag{5}$$

For the power law distribution (Eq 1), α represents the power law exponent. In the exponential distribution (Eq 2), λ represents the rate parameter and is used to indicate the rate of decay. For the log-normal distribution (Eq 3), μ represents the expected value (i.e. mean) and σ represents the standard deviation of the natural logarithm of the variable, x. In the Weibull, or stretched exponential, distribution (Eq 4), λ again represents the rate parameter and β represents the stretching parameter. Finally, for the case of the doubly truncated power law distribution (Eq 5), α and λ are as previously defined in the power law and exponential functions.

In contrast, most, if not all, power law assessments performed in the cardiac science community employed relatively straightforward logarithmic plots of two parameters, e.g. size x and number of events of size x, against each other (i.e. $\log_{10}(y)$ vs $\log_{10}(x)$), accompanied by linear regression of the data. The underlying logic of this method is that if the data fit well to a line on a log-log plot, then the data must obey a power law, because a power law produces a straight line on logarithmic axes [49–51]. However, this is not always true and there could be other, superior descriptors of the data, e.g. exponential distributions or one of the aforementioned heavy-tailed distributions, that this method seldom considers. Consequently, this methodology must be considered incomplete.

This study seeks to demonstrate how analysis conducted using the incomplete methods described above can lead to misleading or contradictory interpretations for a system that demonstrates heavy-tailed, potentially power law behavior. A thorough comparison is made using more robust, established techniques, including: MLE to calculate the (suspected) power law's exponent and log-likelihood ratio tests to engage in comparisons between alternative distributions to which the data could belong [42, 52]. This analysis is applied to the quiescent (stable) period between pacemaker translocations, defined here as the spatial instability and subsequent relocation of the pacemaker region across consecutive beats, as observed in stem cell-derived cardiomyocyte cultures. Pacemaker translocations were observed previously but were not examined in detail [8]. The quiescent periods between pacemaker translocations are suspected to obey a power law due to their superficial similarity to a known power law-obeying system: the inter-burst (or inter-event) interval between neuronal action potential spiking events observed in neuronal cultures [53, 54]. Pacemaker translocation quiescent periods are investigated electrophysiologically using two-dimensional monolayers of stem cell-derived

cardiomyocytes plated onto microelectrode arrays (MEAs). This experimental design provides the requisite spatiotemporal information essential for the analysis of pacemaker translocation quiescent periods in order to determine whether they constitute a power law-adhering phenomenon.

## Materials and methods

### Cell cultures & microelectrode array measurements

Human ESCs were grown and differentiated in a chemically defined condition as previously described [8, 55, 56]. Usage of all the human embryonic stem cell lines is approved by the UCLA Embryonic Stem Cell Research Oversight (ESCRO) Committee and the Institutional Review Boards (IRB) (approval #2009-006-04). Differentiation efficiency is checked periodically by flow cytometry and maintained around 80–90%. Thus, the contamination of non-cardiomyocytes is minimal with these differentiation methods. Two weeks after differentiation, cardiomyocytes were plated, without performing any extra purification step, as two-dimensional monolayers on matrigel-coated (Corning #354277), commercially available microelectrode arrays (MEAs) containing 120 integrated TiN electrodes. All electrodes were 30 μm in diameter with an interelectrode spacing of 200 μm (Multichannel Systems, Reutlingen, Germany). The MEAs were placed in an incubator set to a temperature of 37 ˚C and gas flow of 5% $CO_2$. The cell cultures were given a minimum of 24 hours to ensure the cardiomyocytes were well-attached to each MEA. Local field potentials at each electrode were recorded over an average period of 19.5 minutes, up to twice daily, with a sampling rate of 1 KHz using the MEA2100-HS120 system (Multichannel Systems, Reutlingen, Germany). The full data acquisition yielded 30 MEA recordings across 3 distinct cardiomyocyte cell cultures.

### Computational tools

A custom developed, graphical user interface-based (GUI) Python program, in combination with the powerlaw Python library, was used to conduct the analysis [48]. The program is written for Python 3.8 or above and utilizes a variety of freely available Python libraries, including NumPy, SciPy, Pandas, Matplotlib, Seaborn, Numba, and Statsmodels, among others. The GUI was constructed using PyQt5. The complete code of the GUI program will be made available in a publicly accessible repository.

### Beat detection & pacemaker determination

Beat detection was achieved using the findpeaks function in the SciPy signal processing (scipy. signal) Python library. The key parameters used to identify the peak corresponding to the R-wave in the field potential signal were minimum peak height (electrical amplitude) and minimum peak distance (the allowed spacing between peaks, represented in this case as some time interval). A signal-to-noise ratio in excess of 3:1 was enforced for all potential signals. Furthermore, the detected beats were manually inspected to ensure that the data set parameters correctly identified the R-wave peak for each beat recorded by each electrode. This was done to ensure that the findpeaks function worked as expected.

The pacemaker region of each beat was calculated by identifying the time associated with each R-wave peak recorded for each electrode. Time points were then scaled on a per-beat basis by subtracting the minimum raw time from each time point across all viable electrodes in the beat window. This was done to ensure each beat had a true "zero" value across the array and thereby standardize the data for visualization via heatmap. Therefore, being the

progenitor of the electrical impulse directing a rhythmic beat across a culture, the pacemaker region, or closest proxy for the region, in the array corresponds to the "zero" time point of the given beat.

## Pacemaker translocation algorithm

Pacemaker translocations were detected by monitoring the movement of the pacemaker region across subsequent beats. If the pacemaker region, while moving from one location to another, exceeds a distance threshold (500 μm), then a 'clock' is engaged that counts the number of beats and the duration that the pacemaker region remains in this new location. If the location changes again, the 'clock' is stopped, the number of beats and duration are stored in a list, and the 'clock' is reset for the new position. This process is repeated for each detected pacemaker region of each beat over the full length of the MEA recording. At the end of the calculation, the first recorded event is dropped from the list due to uncertainty regarding how long the pacemaker was in the region prior to the start of the recording. Likewise, the end of the recording does not contribute to an event designation and is similarly not considered. The algorithm was applied both manually through visual inspection and computationally in Python in order to ensure agreement of results.

## Power law analysis & distribution comparisons

Simple or incomplete power law analysis consisted of log-log plots of one variable, X, corresponding to the quiescent period length (the number of beats between translocation events), and Y, corresponding to the number of translocation events (events with a quiescent period of size X). Only events whose quiescent period length occurred more than once (Y > 1) were considered for analysis. This was done in part to eliminate potential outliers within a limited data set. The data were then evaluated using either linear regression or nonlinear least squares fitting of the data to a power law function of the form $F(x) = Ax^{-\alpha}$, where A is a normalization constant and $\alpha$ is the power law exponent. Statistical assessment of the resultant fit, e.g. correlation coefficients and goodness-of-fit metrics, were generated programmatically using established statistical methods in combination with the statsmodels Python library.

Robust power law analysis was performed in accordance with previously described methods and the powerlaw Python library [42, 48]. Visualization was achieved using histograms of pacemaker translocation data and fitting multiple heavy-tailed probability density functions (PDFs), including exponential, power law, log-normal, and Weibull distributions, to the data. Calculation of the power law exponent was achieved using the MLE approach implemented in powerlaw's Fit method with the optional method argument 'discrete' set to 'True'. Log-log plots of both PDFs and complementary cumulative distribution functions (CCDFs) were produced using two methods within the powerlaw library (denoted as object. distribution.plot_pdf() and object.distribution.plot_ccdf() in the library's code) for qualitative comparisons of empirical data-to-distribution fitting. Finally, comparisons between these probability distributions, as well as the doubly truncated power law distribution, were performed using powerlaw's distribution comparison method (denoted as distribution_- compare() in the library's code) with the optional method argument normalized ratio (denoted as normalized_ratio in the powerlaw documentation) set to 'True'. The doubly truncated power law distribution uses an optional argument, $x_{max}$, to define the exponential cutoff. For this experiment, $x_{max} = 150$.

## Results

### Assessment of cardiomyocyte culture viability on MEAs

Microelectrode arrays were used to record cardiomyocyte local field potentials (hereafter called field potentials), which are extracellular electrical signals generated in part by transient imbalances in ion concentrations in the intercellular space [57–59]. These field potentials resemble, but are not equivalent to, electrocardiogram recordings, preventing analysis of P, Q, and S waves. The field potentials were analyzed for each channel (electrode) of each 120 electrode MEA, shown in Fig 1A, across all recordings. These data represent 3 unique cultures of two-dimensional (2D) stem cell-derived cardiomyocyte monolayers, as shown in Fig 1B. The use of 2D monolayers allows for the acquisition of spatially defined information and is further complemented by the precisely manufactured electrode configuration of the recording field of the MEA. Each recording was visually inspected to validate the peak detection methodology for their respective peak amplitude and distance parameters. Except in cases of extreme noise or faulty channels (e.g. due to poor cell-electrode contact, heterogeneous composition of the adhesion layer (Matrigel), or physical degradation of the electrode), peak detection performed as expected. Some representative field potential examples with annotated beat amplitudes (vertical arrow line, gray) and R-wave-like peaks (red x) are shown in Fig 1C and 1D. The average beat rate (beats per minute, bpm) across all datasets was 36.64 bpm with a standard deviation of 14.48 bpm.

### Detection of pacemaker translocations

Beat occurrence times were calculated, normalized to a global maximum, and examined for each beat. These calculations yielded values for the time lag, $t_{lag}$, which is the time taken from when a beat is first detected (pacemaker region, $t_{lag} = 0$ ms) until the last detected peak of the beat ($t_{lag} = t$). Time lag values were assessed for each beat in each recording via heatmap in

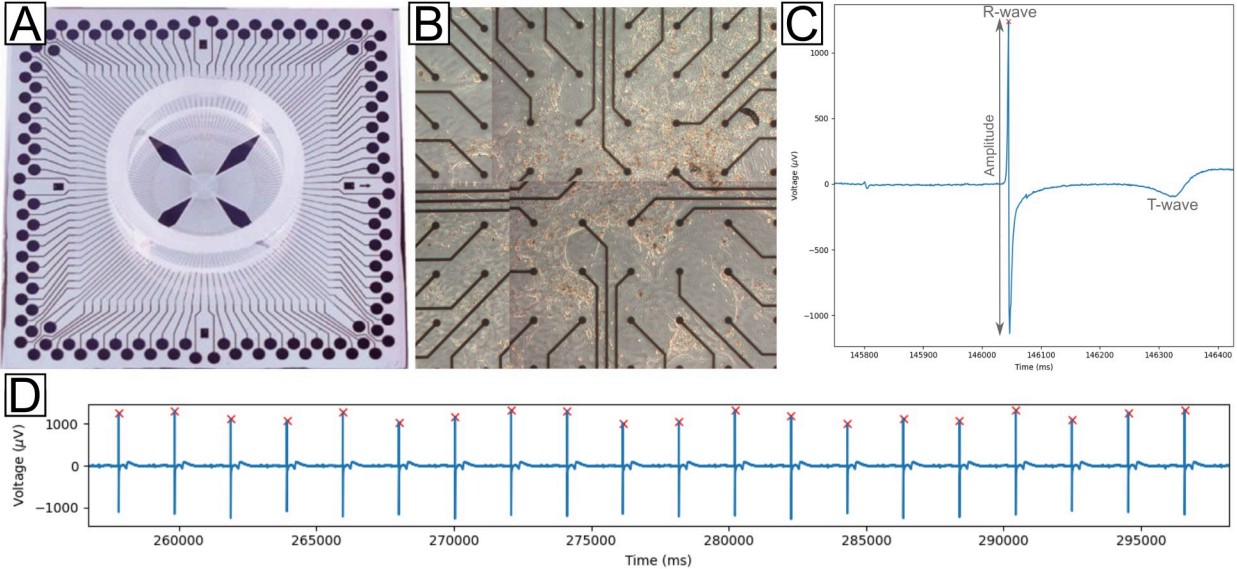

**Fig 1. Representative MEA device, culture and field potential signals.** A) 120 electrode MEA with TiN electrodes. B) Cardiomyocyte culture on a 120 electrode MEA. C) Field potential showing three beats. Beat amplitude is indicated with a vertical arrow line in red. D) Extended view of field potentials for one electrode across a 45 second interval. In both C and D, R-wave-like field potential peaks, as detected by the findpeaks algorithm, are marked with a red X.

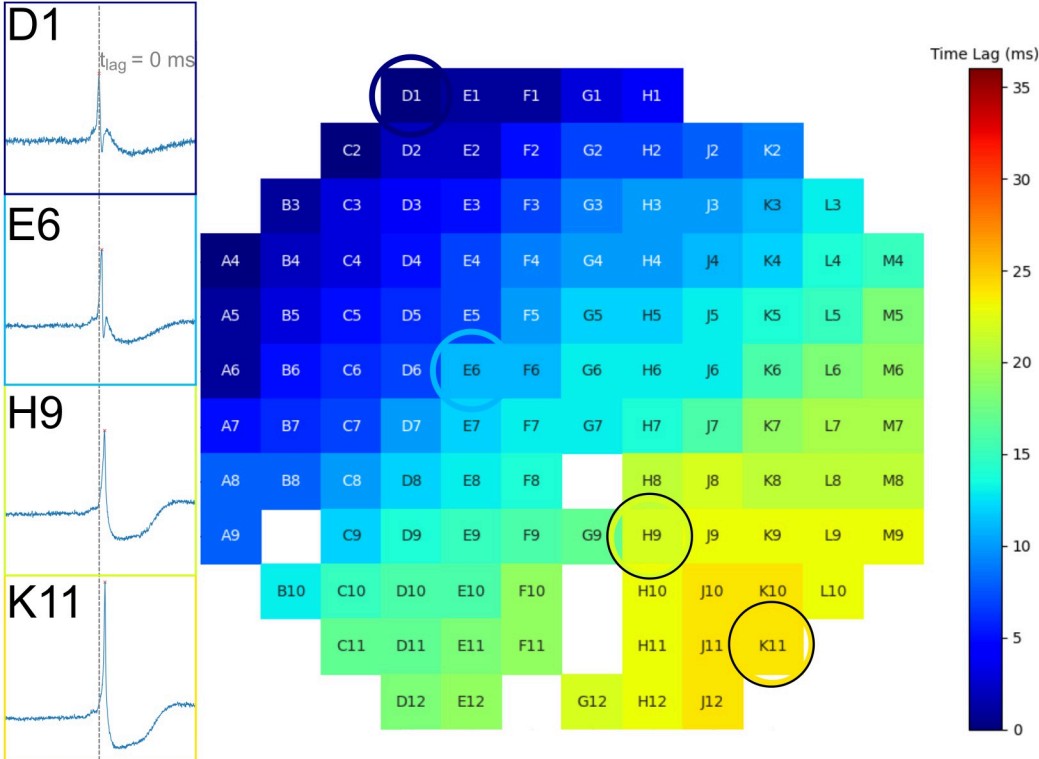

**Fig 2. Field potentials of electrodes in four different regions of the pacemaker map.** The pacemaker region is near electrode D1, which has a recorded normalized time lag of $t_{lag} = 0$ ms. Electrodes further from the pacemaker exhibit a shift to the right of the $t_{lag} = 0$ ms time point (dashed, vertical line) and show how the time at which the signal is detected increases with increasing distance away from the pacemaker region. This shift is correlated with the heatmap time lag values. White squares indicate electrodes that were excluded from analysis due to insufficient signal.

order to view pacemaker regions within the electrode array, as shown in Fig 2. Comparisons between the field potentials and heatmap were consistent: as $t_{lag}$ increased from the minimum (dark blue) to maximum (dark red), there was an observable shift in the field potential relative to $t_{lag} = 0$ ms. Electrodes which did not exhibit detectable beat activity or experienced excessive noise were omitted and colored white. Cursory examinations of field potentials across electrodes failed to reveal any noteworthy changes in signal morphology. Pacemaker regions were observed as frequently lacking a fixed location. The pacemaker region demonstrated instability and moved at least twice in 21 of 30 recordings, while the remaining 9 recordings did not. The movement of the pacemaker region from one location to another is referred to as a pacemaker translocation. Examples of two such translocations are illustrated in Fig 3.

Fig 3A shows the pacemaker region situated closest to the upper left quadrant (dark blue) of the MEA during beat 244. The next beat, 245, is shown in Fig 3B and was observed in a state of flux as the pacemaker region translocated, or moved, from the upper left quadrant to the lower right quadrant (dark blue regions in each respective quadrant, lighter blue in between). The pacemaker region subsequently settled in the lower right quadrant in beat 246, as shown in Fig 3C. This region remained in a stable location from beat 247 to beat 249, which strongly indicated that the translocation phenomenon was due to conditions in the culture (i.e. caused by the cells) and not due to the equipment or analysis program. A second translocation with a clear transitional state is shown from the same recording approximately 308 beats later in Fig

3D–3F. Additional translocations were detected between the translocations depicted in Fig 3, although they are not shown here.

## Non-robust power law analysis of pacemaker translocation quiescent periods

The quiescent period, or the time (measured in beats) between pacemaker translocation events, was calculated using the pacemaker translocation algorithm described earlier. The number of occurrences of a given quiescent period (referred to here as the number of events) were subsequently calculated. The plots shown in Fig 4 constitute the non-robust or "incomplete" analysis of the data. First, a log-log plot of the number of events (y) vs the quiescent period (x) was generated from the data. Linear regression was then applied to the data to produce the plot shown in Fig 4A. On logarithmic axes, the fitted line (black) represents a power law with calculated values for the power law exponent, α = -1.539, and goodness-of-fit, $R^2$ = 0.955. Additional analysis was performed using nonlinear least squares to fit the data to a power law function, as shown in Fig 4B. Here, the calculations yielded α = -1.951 and $R^2$ = 0.995. Finally, the nonlinear least squares fit was plotted on logarithmic axes to produce the plot shown in Fig 4C.

## Robust power law analysis of pacemaker translocation quiescent periods

A robust approach to power law analysis is demonstrated in Fig 5. Fig 5A provides a qualitative assessment using a histogram of the data and probability distribution curves (blue: power law,

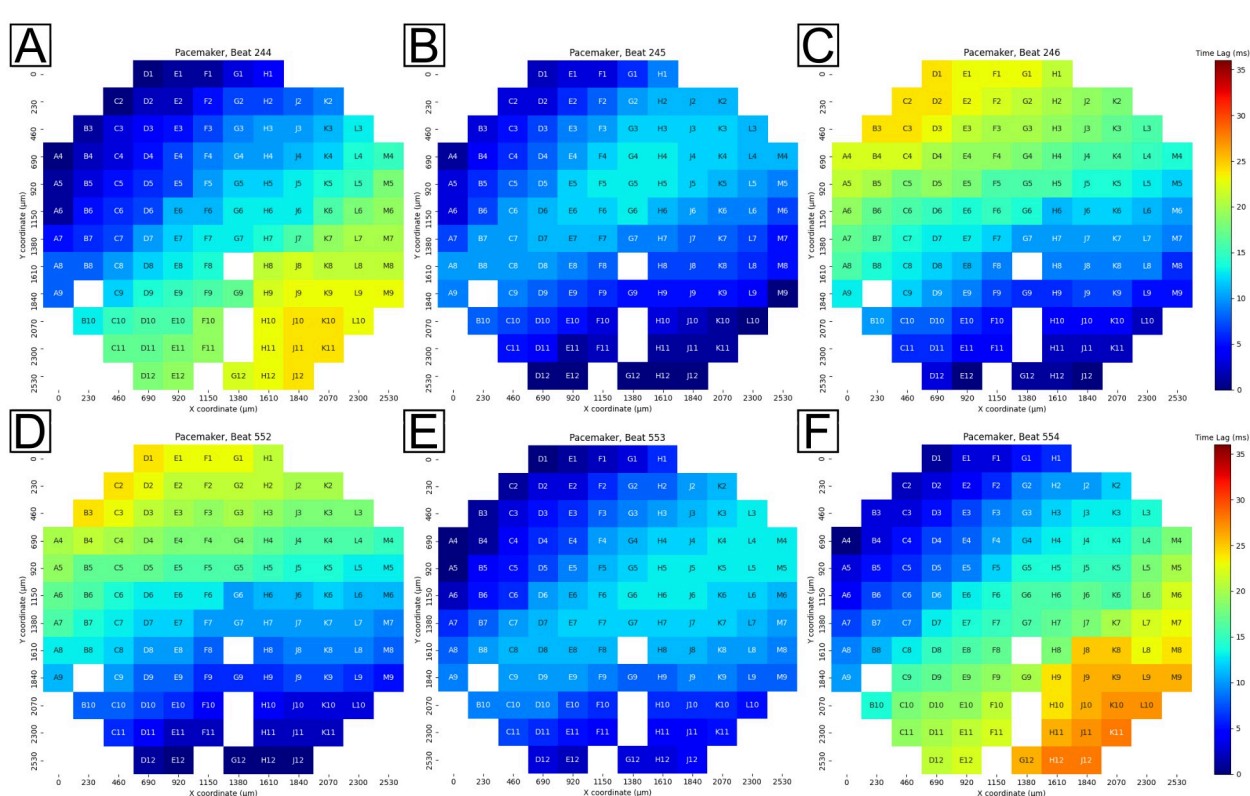

**Fig 3. Example pacemaker translocations observed in one recording of a 120 electrode MEA.** Color bar is normalized from the lowest time lag (0 milliseconds, dark blue) to highest time lag (35 milliseconds, dark red) in the full recording. Top row: the pacemaker region translocates from the top-left corner of the array (A) to a temporary regime positioned simultaneously in the top-left and bottom-right corners (B) before stabilizing in the bottom-right corner of the array (C). Bottom row: another translocation event observed in the same recording over 300 beats later that follows a similar pattern as A-C. The pacemaker is stable in the bottom-right (D) until translocating to a temporary regime positioned simultaneously in the top-left and bottom-right corners (E) before settling in the top-left corner of the array (F).

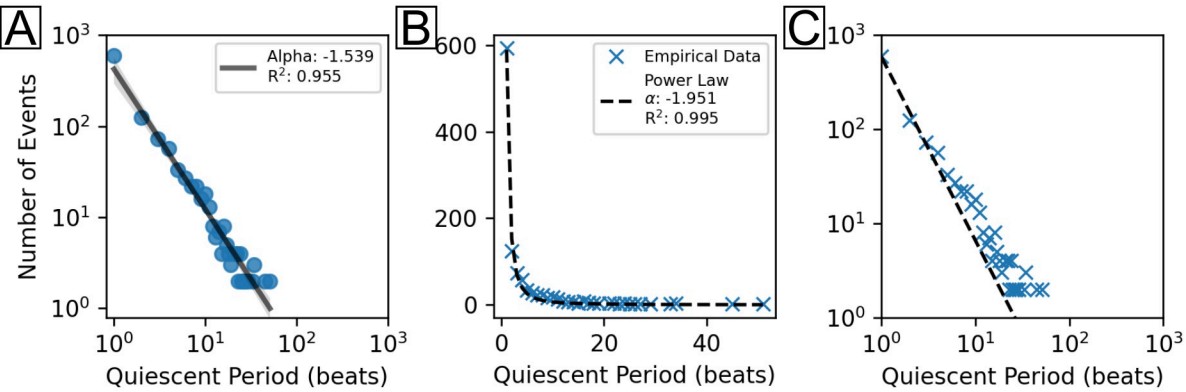

**Fig 4. Non-robust power law analysis.** A) $Log_{10}(y)$ vs $log_{10}(x)$ (scatterplot, blue), where y is the number of events and x is the quiescent period between translocations as measured in the number of beats between. Linear regression of the data (solid line, black) yielded $\alpha = -1.539$ and $R^2 = 0.955$. The confidence interval (0.95) of the linear regression fit is shown in gray shading. Number of unique data points, N: 30. B) Nonlinear least squares fit to a power law function (dashed, black) with the calculated $\alpha = -1.951$ and $R^2 = 0.995$. C) Log-log axis visualization of B.

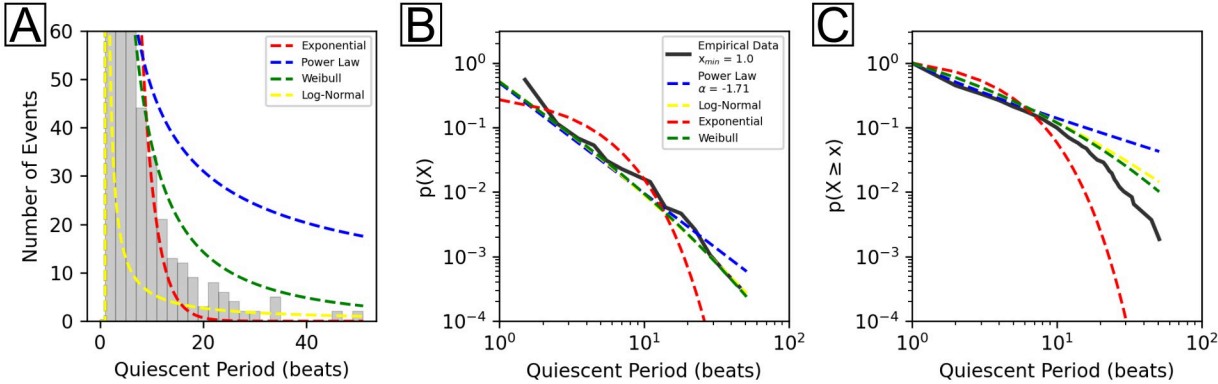

**Fig 5. Comparisons between heavy-tailed probability distributions.** A) Histograms and fitted PDFs of power law (blue, solid), log-normal (yellow, dashed line), exponential (red, dashed line), and Weibull (green, dashed line) probability distributions for all quiescent periods. B) Empirical data (black, solid line) plot overlaid with PDFs of power law (blue, dashed line), log-normal (yellow, dashed line), exponential (red, dashed line) and Weibull (green, dashed line) distributions. C) Empirical data plot overlaid with complementary cumulative distribution functions (CCDFs) of power law, log-normal, exponential, and Weibull distributions. The empirical $x_{min}$, or minimum cutoff value, value was determined algorithmically by the powerlaw library. Calculated parameters for each distribution are summarized in Table 1. Residual plots for the data from B and C are shown in S1 Fig.

red: exponential, yellow: log-normal, green: Weibull) to demonstrate how well the data adhere to each distribution. Here, it is observed that the power law is an inferior fit to the data compared to the Weibull distribution, as judged by the distance between the data and distribution curve. Next, log-log plots of the PDF vs quiescent period and the complementary cumulative distribution function (CCDF) vs quiescent period for each distribution were generated. The interpretation is relatively straightforward: the closer a distribution curve is to the empirical curve, the better the fit between data and distribution. Fig 5B shows the log-log plot of the PDF vs quiescent period for the empirical data (black) and all candidate distributions: power law (blue), log-normal (yellow), exponential (red), and Weibull (green). Table 1 summarizes the parameters calculated for each distribution. The minimum cutoff value, $x_{min}$, was calculated by the powerlaw library ($x_{min} = 1$) as the value which yields an optimal power law fit and is

**Table 1. Summary of calculated parameters for each distribution.**

| Parameter | Distribution | | | | |
|---|---|---|---|---|---|
| | Power Law | Doubly Truncated Power Law | Log-Normal | Exponential | Weibull |
| $\alpha$ | 1.814[a] | 1.583[a] | N/A | N/A | N/A |
| $x_{min}$ | 1.0 | 1.0 | 1.0 | 1.0 | 1.0 |
| $\mu$ | N/A | N/A | 2.363[a] | N/A | N/A |
| $\sigma$ | N/A | N/A | 2.252 | N/A | N/A |
| $\lambda$ | N/A | 0.032 | N/A | 0.315 | 83.884 |
| $\beta$ | N/A | N/A | N/A | N/A | 0.238 |

[a]Value represents the magnitude and omits the sign (which is negative).

required in order to avoid undefined behavior in power law analysis. Through the first decade, the data adhere reasonably well to power law, log-normal, and Weibull distributions but do not follow the exponential distribution. Between the first and second decades, the tail of the data appears to deviate from a power law and instead demonstrates a closer fit to either a log-normal or Weibull distribution. Similarly, Fig 5C shows the CCDF vs quiescent period for the empirical data and all candidate distributions. Here, the deviation of the data from power law behavior is made more explicit as judged by the growing distance between the empirical curve and the power law fit as they diverge at the tail. Notably, the empirical data approaches a vertical limit or asymptote in a manner similar to the exponential distribution as it terminates. A vertical asymptote can be an indication of an exponentially truncated, or doubly truncated, power law. This distribution is evaluated later in the final phase of analysis.

The final phase of analysis engaged in comparisons between candidate distributions using the powerlaw library's distribution comparison function (distribution_compare()). In each comparison, either a power law or doubly truncated power law distribution was selected as the first distribution (numerator) and the alternative heavy-tailed distribution (log-normal, exponential, Weibull) was used for the second distribution (denominator). Log-likelihood ratios (LLRs) and p-values to assess significance of the ratio's sign were calculated by the powerlaw library. If the sign is positive, the first distribution is the more likely fit, and if the sign is negative, the second distribution is the more likely fit. All p-values were compared to the 0.05 significance level ($p \leq 0.05$). A complete summary of the distribution comparison outputs are shown in Table 2.

## Discussion

### Power laws in cardiological systems

One can intuitively surmise that as cardiomyocytes mature, they would exhibit less pacemaker region volatility and more consistent beating. These attributes would be congruous with the establishment of a stable or even permanent pacemaker region such as the sinoatrial node in the heart. Pacemaker instability is already known to be detrimental to heart function: severe abnormalities in pacemaker function necessitate the surgical implantation of electrical devices to maintain heart rhythm [60–62]. Thus, identifying the mechanisms for and improving understanding of pacemaker abnormalities in cardiomyocytes could provide insight into problems afflicting the sinoatrial node of the heart.

Previous studies have shown that there is significant diagnostic power to be found in power law exponents. Investigations of heart rate variability (HRV, also referred to as beat rate variability or BRV), defined as small variations in the interval between heart beats, reported

**Table 2. Log-likelihood ratio (LLR) comparisons between distributions.**

| Parameter | Distribution | |
|---|---|---|
| | **Power Law** | **Doubly Truncated Power Law** |
| | vs. Log-Normal | |
| LLR | -4.043 | 3.806 |
| p-value* | $5.271 \times 10^{-5}$ | $1.412 \times 10^{-4}$ |
| | vs. Exponential | |
| LLR | 8.888 | 10.544 |
| p-value* | $6.194 \times 10^{-19}$ | $5.422 \times 10^{-26}$ |
| | vs. Weibull | |
| LLR | -3.909 | 3.881 |
| p-value* | $9.268 \times 10^{-5}$ | $1.041 \times 10^{-4}$ |
| | vs. Doubly Truncated Power Law | vs. Power Law |
| LLR | -4.049 | 4.049 |
| p-value* | $4.720 \times 10^{-8}$ | $4.720 \times 10^{-8}$ |

*p-values denote the statistical significance of the sign of the LLR. Significance level: 0.05.

differences in power law exponents between healthy, functional tissue and abnormal tissues, e.g. following myocardial infarction or heart transplantation [28, 38–41]. These studies suggest that power law exponents may hold strong predictive power in diagnosing heart abnormalities. However, the use of power laws as diagnostic tools can only be effective if optimal, comprehensive methodologies are applied.

## Suboptimal or incomplete methods lead to misleading interpretations of power law behavior

Suboptimal and optimal power law analysis were performed as shown in Figs 4 and 5, respectively. Fig 4 utilized linear regression of log(y) vs log(x) (Fig 4A) and nonlinear least squares of y vs x on regular (Fig 4B) and logarithmic (Fig 4C) axes. While these methods demonstrated good fits to the data ($R^2$ = 0.955 and 0.995), they were problematic for two key reasons: 1) they did not adequately consider power laws as probability distributions of the form $p(x) \propto x^{-\alpha}$ [63, 64] and 2) they lacked consideration for power law alternatives, e.g. exponential, log-normal and Weibull distributions. These deficiencies were rectified using the methods in Fig 5. Here, qualitative assessments of the fit between data and distributions were conducted using three methods: 1) histogram and PDF curve fits of the quiescent period (Fig 5A), 2) log-log plot of the PDF (Fig 5B) vs quiescent period, and 3) log-log plot of the CCDF (Fig 5C) vs quiescent period.

Fig 5A shows that there is a large gap between the data and the power law fit (blue curve), contradicting the results of Fig 4. Rather, the Weibull distribution (green curve) is, qualitatively, a far better candidate than any of the distributions considered to this point. Thus, the histogram serves as an effective estimation, but is not the preferred qualitative method. Instead, the preferred qualitative assessment for power laws is the log-log plot of the PDF (or, better still, the CCDF) vs event size, as in Fig 5B and 5C. Fig 5B shows that the PDF of the empirical data fits poorly to the power law PDF for most values. The CCDF in Fig 5C shows that the empirical data adhere better to a Weibull distribution (in agreement with Fig 5A) than power law after the first decade. Finally, the gradual decay of the CCDF tail suggests that the data approach a vertical limit or asymptote which could correspond to a maximum cutoff value for the system [48]. Together, these results: 1) thoroughly contradict the interpretation

from Fig 4 that the data obey a power law and 2) indicate the data may fit to another power law alternative: the doubly truncated power law distribution, which is discussed next.

## Pacemaker translocations obey a doubly truncated power law

All power laws are at a minimum singularly truncated, i.e. there is always a minimum value, $x_{min}$ ($x_{min} > 0$), for which power law distributions are valid and, at worst, values below $x_{min}$ could produce undefined behavior. Here, $x_{min}$ was determined algorithmically by the power-law library as previously described [48]. In addition to the lower bound, systems may possess an upper bound, $x_{max}$. The upper bound reflects some natural limitation(s) of the system, e.g. the number of available nodes within a network that is confined to a specific geometric size. This upper bound must be given unique consideration for each system. For a cardiac system, a plausible upper bound is the maximum capacity for the number of beats that a heart (or cardiomyocyte culture) will beat in its lifetime. This is a logical suspicion: because the cardiomyocyte culture cannot beat indefinitely, there could never be a pacemaker translocation quiescent period of infinite length. Under these conditions, it is reasonable to consider truncating the power law at a maximum value using an exponential cutoff. This gives rise to the doubly truncated power law distribution [48, 65, 66]. Evidence to support this interpretation in this experiment was provided by the asymptotic tail observed in Fig 5C. Furthermore, many of the power laws observed in nature are in fact doubly, rather than singularly, truncated [43, 67]. Thus, consideration of a doubly truncated power law distribution is quite reasonable.

Comparisons between power law, doubly truncated power law, exponential, log-normal, and Weibull distributions using log-likelihood ratio (LLR) tests (Table 2) reveal that a power law is disfavored for all cases except the exponential distribution (exponential: LLR = 8.888, log-normal: LLR = -4.043, Weibull: LLR = -3.909, doubly truncated power law: LLR = -4.049; p < 0.05). This result is in agreement with the qualitative assessment from Fig 5 that the power law is a poor descriptor of the data. Equivalent comparisons between doubly truncated power law and alternative distributions favors the doubly truncated power law in all cases (exponential: LLR = 10.544, log-normal: LLR = 3.806, Weibull: LLR = 3.881). Overall, both the robust qualitative and quantitative methods provide strong evidence that pacemaker translocations adhere better to a doubly truncated power law than any of the heavy-tailed distributions considered here.

## Doubly truncated power law exponent indicates potential for a critical system

The doubly truncated power law exponent ($\alpha$ = -1.583), calculated using MLE methods, from Table 2 presents a particularly intriguing possibility: that pacemaker translocations could represent a critical system. Critical systems are systems that demonstrate scale invariant spatiotemporal dynamics, long-range correlations, self-similarity (fractal structures), and power laws, among other features [45, 68–73]. These systems operate at or near a critical point between subcritical ("ordered") and supercritical ("disordered") configurations, analogous to the critical point separating phases of matter in a phase diagram. At the critical point, the system dynamics are markedly different from either the subcritical or supercritical states, leading to the emergence of new properties [72]. Critical dynamics have been observed in both abiotic systems, (e.g. word frequency, earthquake intensity and wildfire frequency) and biotic systems (e.g. animal migration patterns, neurons and the brain) [45, 70, 72, 73]. Importantly, many of these critical systems demonstrated a power law exponent of $\alpha$ = -1.5, which differs by only ~5.5% from the $\alpha$ calculated in this study.

If pacemaker translocations constitute a critical system, then critical dynamics would represent one potential mechanism underpinning the stabilization of the pacemaker region.

Suspected critical systems, e.g. neuronal cultures, have demonstrated optimal information exchange between cooperative units when operating at or near criticality [66, 69, 74–79]. Cardiomyocyte cultures could utilize critical dynamics to determine the optimal location for the pacemaker region to reside. In the cardiomyocyte network, information concerning which cells will become pacemaker cells needs to be conveyed throughout the network across great distances. Operating at or near a critical regime could enable the cardiomyocyte network to maximize signal transduction (i.e. communication of information regarding cellular conditions), ensuring that all cardiomyocytes in the network are in active contact with one another [79–81]. Maximized communication between cardiomyocytes could allow the cells to share structural, local environmental, and genetic information across the entire culture at great distances and speed, which could help determine an optimal region in the culture for the establishment of pacemaker cells. Thus, critical dynamics may play a crucial role in determining the ideal pacemaker region in the cardiomyocyte culture.

## Conclusion

Pacemaker translocations were observed in 21 of 30 cardiomyocyte MEA recordings. A cursory analysis of these translocations using non-robust methods revealed a potential power law relationship between the quiescent period and the number of quiescent periods of a given length. Unfortunately, these techniques are insufficient for assessing power law behavior, partly because they do not represent power laws accurately as probability distributions. More robust methods were subsequently employed to evaluate the data against several probability distributions, including power law, doubly truncated power law, exponential, log-normal, and Weibull. Application of these methods failed to support the interpretation that the data uniquely obey a power law distribution. However, the evidence shows that the data exhibit superior fit to a doubly truncated power law with $\alpha = -1.583$. These findings demonstrate that 1) misleading conclusions are likely if less robust or incomplete methods are applied to power law analysis of cardiological phenomena, 2) pacemaker translocations obey a doubly truncated power law distribution, and 3) indicate the potential for critical dynamics in the establishment of the pacemaker region in cardiomyocyte cultures.

## Supporting information

**S1 Fig. Residual plots calculated for the distribution data depicted in Fig 5B and 5C.** Each residual plot shows the difference between the real data values (empirical PDF or CCDF) and the model values (distribution PDF or CCDF). These residual values are plotted against the quiescent period as measured in beats.
(TIF)

## Acknowledgments

CSD would like to acknowledge the work of Michelle (Nguyen) Phi, at the time an undergraduate biochemistry student at UCLA, in the early stages of this project. Her contributions to the codebase, early analysis, and troubleshooting were essential in pushing this project to its present state.

## Author Contributions

**Conceptualization:** Christopher S. Dunham, Adam Z. Stieg, James K. Gimzewski.

**Data curation:** Christopher S. Dunham, Madelynn E. Mackenzie, Haruko Nakano.

**Formal analysis:** Christopher S. Dunham, Madelynn E. Mackenzie.

**Funding acquisition:** Atsushi Nakano, Adam Z. Stieg, James K. Gimzewski.

**Investigation:** Christopher S. Dunham, Madelynn E. Mackenzie, Alexis R. Kim.

**Methodology:** Christopher S. Dunham, Haruko Nakano.

**Project administration:** Adam Z. Stieg, James K. Gimzewski.

**Resources:** Christopher S. Dunham, Haruko Nakano, Atsushi Nakano, Adam Z. Stieg, James K. Gimzewski.

**Software:** Christopher S. Dunham, Madelynn E. Mackenzie.

**Supervision:** Atsushi Nakano, Adam Z. Stieg, James K. Gimzewski.

**Validation:** Christopher S. Dunham, Madelynn E. Mackenzie, Alexis R. Kim.

**Visualization:** Christopher S. Dunham.

**Writing – original draft:** Christopher S. Dunham.

**Writing – review & editing:** Christopher S. Dunham, Madelynn E. Mackenzie, Haruko Nakano, Alexis R. Kim, Michal B. Juda, Atsushi Nakano, Adam Z. Stieg, James K. Gimzewski.

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
