## [Decision Letter · Decision Letter 0]

20 Jan 2022

PONE-D-21-40812Pacemaker translocations and power laws in 2D stem cell-derived cardiomyocyte culturesPLOS ONE

Dear Dr. Dunham,

Thank you for submitting your manuscript to PLOS ONE. After careful consideration, we feel that it has merit but does not fully meet PLOS ONE’s publication criteria as it currently stands. Therefore, we invite you to submit a revised version of the manuscript that addresses the points raised during the review process.

We look forward to receiving your revised manuscript.

Kind regards,

Xiaoping Bao, Ph.D.

Academic Editor

PLOS ONE

Journal Requirements:

"This work was funded by grants from

the National Institutes of Health (R21HL124503, R01 HL142801 and HL146159). MBJ

was supported by the UCLA Center for Duchenne Muscular Dystrophy Ruth L. Kirschstein

National Research Service Award T32AR065972 ‘Muscle Cell Biology, Pathophysiology,

and Therapeutics’ from the National Institute of Arthritis and Musculoskeletal and Skin

Diseases."

"This work was funded by grants from the National Institutes of Health (https://www.nih.gov/grants-funding). Grant R21HL124503 was awarded to AZS, JKG, and AN. Grants R01 HL142801 and HL146159 were awarded to AN.

MBJ was supported by the UCLA Center for Duchenne Muscular Dystrophy Ruth L. Kirschstein National Research Service Award T32AR065972 ‘Muscle Cell Biology, Pathophysiology, and Therapeutics’ from the National Institute of Arthritis and Musculoskeletal and Skin Diseases (https://grants.nih.gov/grants/guide/pa-files/pa-21-048.html).

Reviewers' comments:

Reviewer's Responses to Questions

**Comments to the Author**

1. Is the manuscript technically sound, and do the data support the conclusions?

Reviewer #1: Yes

Reviewer #2: Yes

2. Has the statistical analysis been performed appropriately and rigorously? 

Reviewer #1: Yes

Reviewer #2: Yes

3. Have the authors made all data underlying the findings in their manuscript fully available?

Reviewer #1: Yes

Reviewer #2: Yes

4. Is the manuscript presented in an intelligible fashion and written in standard English?

Reviewer #1: Yes

Reviewer #2: Yes

5. Review Comments to the Author

Reviewer #1: This paper reports pacemaker translocations occurring in 2-dimensional hPSC-derived cardiomyocyte culture and power law analysis of these translocation events. While it is interesting to see analysis of cardiomyocyte behaviors based on a statistical power-law relation, several aspects of the manuscript can be potentially improved. The detailed comments are as follows:

1. One of the conclusions of the manuscript is that the pacemaker translocations obey a doubly truncated power law distribution. To claim this, the results should be reproduced for multiple differentiations as there can be variabilities in each differentiation (i.e. differentiation efficiency). It is unclear whether the three distinct cardiomyocyte cell cultures are from single differentiation subcultured onto three distinct MEA plates or from three independent differentiations. Parameters obtained from each differentiation can possibly be presented as mean ± SD to give better statistical reliability.

2. The manuscript lacks molecular characterization of hPSC-derived cardiomyocytes. Although beating is one of the hallmarks of cardiomyocytes, cardiomyocyte differentiation typically gives non-myocytes populations in culture which would affect the results. It is important to show the quality of the cells (i.e. cardiomyocyte purity) used in the MEA analysis or at least mention that there can be non-myocyte cells in culture.

3. The manuscript lacks detailed description of cardiomyocyte differentiation (& purification step, if any) protocol. Especially, the time point of cells used for MEA analysis should be mentioned as that can be correlated with the maturation status of the cardiomyocytes.

4. Typical cardiomyocyte differentiation yields ventricular-like cardiomyocytes rather than nodal-like (pacemaker) cardiomyocytes. One might ask if the cells on the pacemaker region have characteristics of nodal-like cardiomyocytes. I wonder if there is a simple way to characterize action potential of the cells using MEA to identify subtype-specific action potential profile of the cells. If not, it would be better to at least mention the subtypes of cardiomyocytes and potential pacemaker function of hPSC-derived cardiomyocytes (generally, hPSC-derived cardiomyocytes consist of mixture of different subtypes even if most of them are ventricular-like cells).

Reviewer #2: The ability to model pacemaker translocation in 2D culture through simple mathematical modeling is vital to correctly understand dynamic behaviors of stem cell-derived cardiomyocytes. Here, Dunham et al. measured pacemaker translocations by microelectrode arrays and compared modeling of these dynamic event by log-normal, exponential, Weibull, and a doubly truncated power law against non-robust methods that only use simple power laws.

While we are enthusiastic about this work, we believe the article can benefit from several content addition and rewording/formatting edits. At times, we found that mathematical equations and estimated parameters were missing for several models. In addition, improved quantitative assessment for model fitting against data should be performed. Furthermore, certain formatting issues are distracting and need to be corrected. We recommend this article for publication after major revision, and request the authors to consider the points below:

Major points:

1. Equations for log-normal, exponential, Weibull, and the doubly truncated power law with data-dependent parameters should be explicitly shown in the paper.

2. To quantitatively assess model fitting, methods such as residual plots should be included in addition to simple visual examinations.

3. Additional literature can be cited to better explain and support the background section of the paper:

1) Seminal works that contribute to stem cell-derived cardiovascular cultures should be discussed and cited in the first paragraph of the introduction when it discusses “have shone significant light on numerous factors contributing to the development of the heart.” Although we understand that cardiomyocytes are the major focus for this study, differentiation methods for other cardiovascular components such as endothelial, vascular smooth muscle cells, etc. should also be mentioned: 10.1038/nprot.2017.033, 10.1016/j.stemcr.2014.09.005, 10.1161/ATVBAHA.117.309196, 10.1016/j.stemcr.2021.04.019, 10.1073/pnas.1200250109, and 10.1038/nmeth.2999.

2) Previous studies for several power-law applications in cardiomyocytes are missing Please consider adding these relevant reports: 10.1016/j.bios.2020.112399, 10.22489/CinC.2017.207-155, and doi.org/10.1161/CIRCEP.116.004508.

4. Table 1 should include parameters for each model. Currently, α and xmin are the only two parameters shown there for simple power law.

5. Figures should be copied with high resolution. At times, many of them look quite blurry.

6. With all due respect, the manuscript contains several formatting issues. Examples include, but not limited to, the paragraph between Figure 5 captions and Table 1. Additionally, the underscore between the word “library’s distribution” and “compare” should be deleted in this paragraph.

Minor points:

1. Equations for power laws should be explained using plain texts or shown as a separate equation item (like Equation X).

2. Transitions should be made between pacemaker cells and power laws (the 2nd and 3rd paragraphs from the introduction section).

3. Latin words such as “in vivo” and “in vitro” should be italicized.

6. PLOS authors have the option to publish the peer review history of their article (what does this mean?). If published, this will include your full peer review and any attached files.

Reviewer #1: No

Reviewer #2: No

---

## [Author Response · Author response to Decision Letter 0]

29 Jan 2022

We have provided a response to each editor and reviewer comment in our Response to Reviewers document. We have copied and pasted the contents of that document into this field below:

Reviewer 1 Comments and Response

This paper reports pacemaker translocations occurring in 2-dimensional hPSC-derived cardiomyocyte culture and power law analysis of these translocation events. While it is interesting to see analysis of cardiomyocyte behaviors based on a statistical power-law relation, several aspects of the manuscript can be potentially improved. The detailed comments are as follows:

1. One of the conclusions of the manuscript is that the pacemaker translocations obey a doubly truncated power law distribution. To claim this, the results should be reproduced for multiple differentiations as there can be variabilities in each differentiation (i.e. differentiation efficiency). It is unclear whether the three distinct cardiomyocyte cell cultures are from single differentiation subcultured onto three distinct MEA plates or from three independent differentiations. Parameters obtained from each differentiation can possibly be presented as mean ± SD to give better statistical reliability.

Thank you for this comment. The data we’ve presented is an aggregate of three distinct differentiations. We evaluated the data both in aggregate and as individual (i.e. grouping the data from a distinct differentiation only), averaged values. We found comparable values for the power law exponent, α, regardless of the approach. However, the limited number of available data points made a meaningful statistical analysis for each distribution challenging. Thus, because the calculated power law exponents were similar and analyzing the data in aggregate avoids these statistical shortfalls during distribution comparisons, we chose to present the data in aggregate.

2. The manuscript lacks molecular characterization of hPSC-derived cardiomyocytes. Although beating is one of the hallmarks of cardiomyocytes, cardiomyocyte differentiation typically gives non-myocytes populations in culture which would affect the results. It is important to show the quality of the cells (i.e. cardiomyocyte purity) used in the MEA analysis or at least mention that there can be non-myocyte cells in culture.

We thank you for pointing this out. The purity of the cardiomyocytes derived from hESC with our differentiation method is approximately over 90 % as a mean value confirmed by flow cytometry for MF20 and cTnT expression (Zhu et al, Sci Rep, 2017; Nakano et al, eLife, 2017). 

We agree with Reviewer 1’s comment on the presence of non-myocyte cells in culture. Although we could not perform the assays for differentiation efficiency for each experiment, we can safely assume that our protocol guarantees the minimal non-myocyte fraction in culture.

The method section was edited according to the comments. In addition to the purity, the further molecular characterization of our hPSC-derived cardiomyocytes was reported in Figure 1 of the paper by Zhu et al (doi: 10.1038/srep43210). 

3. The manuscript lacks detailed description of cardiomyocyte differentiation (& purification step, if any) protocol. Especially, the time point of cells used for MEA analysis should be mentioned as that can be correlated with the maturation status of the cardiomyocytes.

Thank you for the comments. The time point of the cells used for MEA analysis was day 16 in the majority of cases (by starting analysis two days after plating as of day14 differentiation), and day 36 cardiomyocytes in some recordings. The details of our differentiation method is described in the paper referenced in the previous comment 2. In relation to the previous comment, our differentiation method does not employ the purification step due to the high yield of differentiation efficiency. The method section was edited accordingly.

4. Typical cardiomyocyte differentiation yields ventricular-like cardiomyocytes rather than nodal-like (pacemaker) cardiomyocytes. One might ask if the cells on the pacemaker region have characteristics of nodal-like cardiomyocytes. I wonder if there is a simple way to characterize action potential of the cells using MEA to identify subtype-specific action potential profile of the cells. If not, it would be better to at least mention the subtypes of cardiomyocytes and potential pacemaker function of hPSC-derived cardiomyocytes (generally, hPSC-derived cardiomyocytes consist of mixture of different subtypes even if most of them are ventricular-like cells).

We thank Reviewer 1 for his/her insightful comments. Unfortunately, the current MEA systems are not able to identify a subtype of cardiomyocytes. As Reviewer 1 pointed out, the hPSC-derived cardiac differentiation generates a mixture of different subtypes of cardiomyocytes. The article (Minami, 2012 Cell Report) on which our cardiomyocyte differentiation protocol is based characterized the subtypes of the cardiomyocytes generated with this differentiation protocol. This paper reported that this method generates approximately 60% or 8% of MLC2v-positive mature ventricular type cells or MLC2v/MLC2a double-positive immature cells respectively and few MLC2a-positive cells. In addition to the marker expression, they performed a whole-cell patch-clamp assay and identified the functional cardiomyocytes with the action potential property indicating ventricular cells and pacemaker cells. 

Reviewer 2 Comments and Response

The ability to model pacemaker translocation in 2D culture through simple mathematical modeling is vital to correctly understand dynamic behaviors of stem cell-derived cardiomyocytes. Here, Dunham et al. measured pacemaker translocations by microelectrode arrays and compared modeling of these dynamic events by log-normal, exponential, Weibull, and a doubly truncated power law against non-robust methods that only use simple power laws.

While we are enthusiastic about this work, we believe the article can benefit from several content additions and rewording/formatting edits. At times, we found that mathematical equations and estimated parameters were missing for several models. In addition, improved quantitative assessment for model fitting against data should be performed. Furthermore, certain formatting issues are distracting and need to be corrected. We recommend this article for publication after major revision, and request the authors to consider the points below:

Major points:

1. Equations for log-normal, exponential, Weibull, and the doubly truncated power law with data-dependent parameters should be explicitly shown in the paper.

Thank you for this suggestion. We agree that these equations should be shown and have added them to the manuscript, along with additional sentences to describe the meaning of the parameters, as recommended.

2. To quantitatively assess model fitting, methods such as residual plots should be included in addition to simple visual examinations.

Thank you for this suggestion. We have generated residual plots for the PDF and CCDF data from Figs 5B and 5C. We have added this as a supplemental figure, S1 Fig, in a new Supporting Information document, and included a note in the manuscript.

3. Additional literature can be cited to better explain and support the background section of the paper:

1) Seminal works that contribute to stem cell-derived cardiovascular cultures should be discussed and cited in the first paragraph of the introduction when it discusses “have shone significant light on numerous factors contributing to the development of the heart.” Although we understand that cardiomyocytes are the major focus for this study, differentiation methods for other cardiovascular components such as endothelial, vascular smooth muscle cells, etc. should also be mentioned: 10.1038/nprot.2017.033, 10.1016/j.stemcr.2014.09.005, 10.1161/ATVBAHA.117.309196, 10.1016/j.stemcr.2021.04.019, 10.1073/pnas.1200250109, and 10.1038/nmeth.2999.

2) Previous studies for several power-law applications in cardiomyocytes are missing Please consider adding these relevant reports: 10.1016/j.bios.2020.112399, 10.22489/CinC.2017.207-155, and doi.org/10.1161/CIRCEP.116.004508.

These references are greatly appreciated. We’ve elected to incorporate 4 references from the first set and 2 from the second set into the manuscript.

4. Table 1 should include parameters for each model. Currently, α and xmin are the only two parameters shown there for simple power law.

We appreciate this recommendation. We found the cleanest way to implement this suggestion was to split Table 1 into two tables. The new Table 1 now summarizes the model parameters, as suggested. The new Table 2 contains the distribution comparisons between power law and doubly truncated power law versus the alternative distributions, along with the log-likelihood ratios and p-values.

5. Figures should be copied with high resolution. At times, many of them look quite blurry.

Thank you for this notification. We have provided higher resolution versions of each figure for this revised submission.

6. With all due respect, the manuscript contains several formatting issues. Examples include, but not limited to, the paragraph between Figure 5 captions and Table 1. Additionally, the underscore between the word “library’s distribution” and “compare” should be deleted in this paragraph.

Thank you for your comment. We do apologize for submitting this manuscript without utilizing the proper formatting throughout. That was an oversight on our part and has been remedied. We have also fixed the paragraph between Figure 5 captions and Table 1, as you highlighted. Regarding the underscore, we wrote that with the intent to communicate how the function is named in the library, should someone wish to utilize the library and use the same function. In light of your feedback, we’ve decided to rephrase the sentence in question to remove the underscore and provide the version with the underscore as a parenthetical aside. We did the same to instances of “normalized_ratio”.

Minor points:

1. Equations for power laws should be explained using plain texts or shown as a separate equation item (like Equation X).

We agree with this recommendation and have amended the manuscript to list the equations as separate items and describe their parameters.

2. Transitions should be made between pacemaker cells and power laws (the 2nd and 3rd paragraphs from the introduction section).

This comment was greatly appreciated and we agree with the suggestion. We have added such a paragraph to improve the transition from pacemaker development to interest in power laws.

3. Latin words such as “in vivo” and “in vitro” should be italicized.

Thank you for informing us about this error. This issue has been corrected.

List of Reference Removals and Additions

One reference was removed for this revised manuscript due to lack of relevance to our study:

[formerly 19] Mommersteeg MTM, Dominguez JN, Wiese C, Norden J, de Gier-de Vries C, Burch JBE, et al. The sinus venosus progenitors separate and diversify from the first and second heart fields early in development. Cardiovasc Res. 2010;87: 92-101. doi: 10.1093/cvr/cvq033.

Several references were added, including most of those recommended by Reviewer 2. A full accounting of the new references (and their corresponding number in the manuscript) is provided below, as requested:

[3] Breckwoldt K, Letuffe-Brenière D, Mannhardt I, Schulze T, Ulmer B, Werner T, et al. Differentiation of cardiomyocytes and generation of human engineered heart tissue. Nat Protoc. 2017;12: 1177–1197. doi: 10.1038/nprot.2017.033.

[4] Lian X, Hsiao C, Wilson G, Zhu K, Hazeltine LB, Azarin SM, et al. Cardiac differentiation of jPSCs via Wnt Signaling. Proc Natl Acad Sci. 2012;109: E1848-E1857. doi: 10.1073/pnas.1200250109

[5] Maguire EM, Xiao Q, Xu Q. Differentiation and Application of Induced Pluripotent Stem Cell-Derived Vascular Smooth Muscle Cells. Arterioscler Thromb Vasc Biol. 2017;37: 2026–2037. doi: 10.1161/ATVBAHA.117.309196

[6] Xiaojun L, Xiaoping B, Abraham A, Jialu L, Yue W, Dong W, et al. Efficient differentiation of human pluripotent stem cells to endothelial progenitors via small-molecule activation of Wnt signaling. Stem Cell Rep. 2014;3: 804-16. doi: 10.1016/j.stemcr.2014.09.005.

[21] Chakraborty S, Yutzey KE. Tbx20 regulation of cardiac cell proliferation and lineage specialization during embryonic and fetal development in vivo. Dev Biol. 2012;363: 234-246. doi: 10.1016/j.ydbio.2011.12.034

[22] Yutzey KE. Cardiomyocyte proliferation: teaching an old dogma new tricks. Circ Res. 2017;120: 627–629. doi: 10.1161/CIRCRESAHA.116.310058

[23] Ribeiro da Silva A, Neri EA, Turaça LT, Dariolli R, Fonseca-Alaniz MH, Santos-Miranda A, et al. NOTCH1 is critical for fibroblast-mediated induction of cardiomyocyte specialization into ventricular conduction system-like cells in vitro. Sci Rep. 2020;10. doi: 10.1038/s41598-020-73159-0

[24] Mandla R, Jung C, Vedantham V. Transcriptional and epigenetic landscape of cardiac pacemaker cells: insights into cellular specialization in the sinoatrial node. Front Physiol. 2021;12. doi: 10.3389/fphys.2021.712666

[36] Kim J, Kuusela J, Aalto-Setala K, Rasanen E. Short- and long-range correlations in beat rate variability of human pluripotent-stem-cell-derived cardiomyocytes. Comput. Cardiol. 2017;44. doi: 10.22489/CinC.2017.207-155

[37] Wang X, Wang L, Dou W, Huang Z, Zhao Q, Mahli M, Maynes JT, Sun Y. Electrical impedance-based contractile stress measurement of human iPSC-Cardiomyocytes. Biosens. Bioelectron. 2020;166: 112399. doi: 10.1016/j.bios.2020.112399

---

## [Decision Letter · Decision Letter 1]

2 Feb 2022

Pacemaker translocations and power laws in 2D stem cell-derived cardiomyocyte cultures

PONE-D-21-40812R1

Dear Dr. Dunham,

We’re pleased to inform you that your manuscript has been judged scientifically suitable for publication and will be formally accepted for publication once it meets all outstanding technical requirements.

Kind regards,

Xiaoping Bao, Ph.D.

Academic Editor

PLOS ONE

Additional Editor Comments (optional):

Reviewers' comments:

Reviewer's Responses to Questions

**Comments to the Author**

1. If the authors have adequately addressed your comments raised in a previous round of review and you feel that this manuscript is now acceptable for publication, you may indicate that here to bypass the “Comments to the Author” section, enter your conflict of interest statement in the “Confidential to Editor” section, and submit your "Accept" recommendation.

Reviewer #1: All comments have been addressed

Reviewer #2: All comments have been addressed

2. Is the manuscript technically sound, and do the data support the conclusions?

Reviewer #1: Yes

Reviewer #2: Yes

3. Has the statistical analysis been performed appropriately and rigorously? 

Reviewer #1: Yes

Reviewer #2: Yes

4. Have the authors made all data underlying the findings in their manuscript fully available?

Reviewer #1: Yes

Reviewer #2: Yes

5. Is the manuscript presented in an intelligible fashion and written in standard English?

Reviewer #1: Yes

Reviewer #2: Yes

6. Review Comments to the Author

Reviewer #1: I appreciate that the authors addressed all the comments and made relavent changes in the manuscript.

Reviewer #2: The authors have satisfactorily addressed all my questions and comments. I hereby recommend acceptance of this paper.

7. PLOS authors have the option to publish the peer review history of their article (what does this mean?). If published, this will include your full peer review and any attached files.

Reviewer #1: No

Reviewer #2: No

---

## [Editor Report · Acceptance letter]

4 Mar 2022

PONE-D-21-40812R1 

Pacemaker translocations and power laws in 2D stem cell-derived cardiomyocyte cultures 

Dear Dr. Dunham:

I'm pleased to inform you that your manuscript has been deemed suitable for publication in PLOS ONE. Congratulations! Your manuscript is now with our production department. 

Kind regards, 

on behalf of

Dr. Xiaoping Bao 

Academic Editor

PLOS ONE